# Analysis of Load-Sharing Behavior of the Multistage Planetary Gear Train Used in Wind Generators: Effects of Random Wind Load

**Jungang Wang [1],\*, Shinan Yang [1], Yande Liu [1] and Ruina Mo [2]**

[1] School of Mechanotronics and Vehicle Engineering, East China Jiaotong University, Nanchang 330013, China; 17862230711@163.com (S.Y.); 2607@ecjtu.edu.cn (Y.L.)

[2] School of Basic Science, East China Jiaotong University, Nanchang 330013, China; mrn810211@163.com

\* Correspondence: sduhys@163.com

**Abstract:** Load-sharing behavior is very important for power-split gearing systems. Taking the multistage planetary gear train transmission of an Million Watt (MW) wind generator as the investigation object, and based on the gear transmission system of a wind generator in a complex and changing load environment, a random wind model of a wind farm is built by using the two-parameter Weibull distribution. According to the realistic working region of the wind generator, the random wind speed is changed into time-varying input speed of the wind generator gear box. Considering the internal excitation, such as mesh stiffness, mesh damping of gear pairs and the meshing error, a dynamic model for a multistage planetary gear transmission system is built by using the lumped parameter method. The load-sharing coefficients are obtained for each planet gear pair in the same meshing period of the transmission system that is under the interaction of time-varying input speed and internal excitation. It is shown that the degree of load-sharing coefficient fluctuation for each planet gear pair of the first- and second-stage planetary gear train is significantly affected by time-varying input speed. The research results can lay a theoretical foundation for optimization and reliability of planetary gear transmission systems of wind generators.

**Keywords:** wind generator; multistage planetary gear train; random wind speed; load sharing

## 1. Introduction

The planetary gear transmission has advantages of a large transmitting ratio, compact structure and strong load-bearing capacity, and has been widely applied in wind turbine gearboxes. However, the wind turbine gearbox is usually installed at the wind gap of mountains, islands, etc., and it is subjected to an irregular wind force year-round. Under the working conditions of violent gusts, the load distribution of planet gears is more uneven; this will cause much vibration and noise in the transmission system. More severely, the wind turbine gearbox will fail to work if the load distribution is mostly concentrated on one planet gear. Therefore, it is significant to analyze the load-sharing characteristics of the transmission system of a wind turbine gearbox to ensure its operating reliability and for improving its service life.

For a long time, scholars at home and abroad have focused on the load-sharing behavior of the planetary gear transmission system. Kahraman [1] explored the load-sharing behavior of planetary gear transmissions from the perspective of statics. Using experiments, Ligata et al. [2] found that the position error of the planet gear was a key factor that affected load sharing of a transmission system.

Singh [3] analyzed the influence of the number of planet gears on the load-sharing characteristics of a planetary gear transmission. From the perspective of statics, Lu et al. [4] and Tang et al. [5] analyzed

the relationship between position error and the load-sharing characteristics of an NGW (N: internal gear pair; W: external gear pair; G: a common gear engaged with two central gears simultaneously) planetary gear transmission. Gu and Velex [6] presented an original lumped parameter model of a planetary gear transmission to account for planet gear position errors and simulated their contribution to the dynamic load sharing among the planet gears. Sheng et al. [7] proposed a double-row planetary gear set and analyzed the behavior of static-load-sharing characteristics affected by gear-eccentricity error, ring gear supporting stiffness, planet bearing stiffness and torsional stiffness of the first-stage carrier. Wu et al. [8] established a non-linear dynamics model of compound planetary gear sets based on the lumped parameter theory and analyzed the influence of position and eccentric errors on load-sharing characteristics of the gear sets. Wang et al. [9,10] focused on the multistage planetary gear transmission system, established a dynamic model based on lumped parameter theory and analyzed the influence of revolution speed and mesh error on the dynamic load-sharing characteristics of the gear sets. Zhu et al. [11] established a dynamic model of the planetary gear train system by considering time-varying mesh stiffness, revolution damping and component gravity and studied the impact of support stiffness on the dynamic load-sharing characteristics of the transmission system. Chen et al. [12] developed a purely rotational non-linear dynamic model of a planetary gear transmission system by considering time-varying mesh stiffness and synthetic mesh errors and researched the dynamic behavior of the wind turbine with the random fluctuation of synthetic mesh errors. Qin et al. [13] studied the vibration speed, displacement of gears and mesh force of each gear of the wind generator transmission system under the interaction of time-varying input torque and internal excitation. To evaluate non-equilibrium effects, Yi et al. [14] established a coupled dynamic of the planetary gear system in consideration of nonlinear factors and found that static load sharing is conservative. Shi et al. [15] established a dynamic-model wind turbine planetary gear transmission system by using a lumped-parameter method to consider the internal excitation, and analyzed the vibration displacement and vibration velocity of components in the transmission system under varying wind load. Mo et al. [16] built a refined mathematical model for a star-gearing reducer in consideration of different mesh stiffness and supporting stiffness for components and analyzed the load-sharing characteristics of the planet gear under the influence of interactions and various component errors. Dong et al. [17] presented a nonlinear dynamic model of translation-torsion for a single-stage planetary gear train with three planet gears and analyzed the load-sharing characteristics of the planet gear under support stiffness, manufacture and assembly errors. Jing et al. [18] studied the fatigue life of a sun gear in a wind turbine gearbox based on the fatigue-life prediction model of components under stochastic loading. Zhao et al. [19] established a pure-torsion coupling, nonlinear dynamic model of a Million Watt (MW)-level wind power generator, and researched the response of angular displacement, angular speed and gearing force of torsional vibration of the components in the transmission train system under random wind load. Kim et al. [20] studied the effects of increasing torque and changing rotation direction of the planetary gearbox on the dynamic load-sharing among the planet gears.

All these researches had important effects on the planetary gear transmission system but targeted other transmission types and focused on different fields, most of them are not involving the study of load-sharing characteristics of a multistage planetary gear transmission system under random wind load. In this study, a transmission scheme of a multistage gear train composed of a two-stage planetary gear train and a one-stage parallel-axis gear is the study objective, and the dynamic model is established on the basis of lumped parameter theory. By using a numerical method, the dynamic load-sharing coefficients of each gear pair in the same meshing period of the planetary gear transmission system are analyzed under the interaction of time-varying input speed and internal excitation.

## 2. Random Wind-Speed Model Based on Weibull Distribution

It is very important to know how to simulate the random wind-speed model for studying the load-sharing characteristics of a planetary gear transmission system. This is because the input speed of

the wind generator gearbox is determined by the random wind speed in a wind farm. The Weibull distribution is a two-parameter function commonly used to fit the wind-speed frequency distribution. The Weibull function provides a convenient representation of wind-speed data for wind-energy calculation purposes, and the function is commonly referred to as the wind-speed distribution. The probability density function of the Weibull distribution is given by:

$$f(v) = \frac{kv^{k-1}}{c^k} \exp\left[-\left(\frac{v}{c}\right)^k\right],$$ (1)

where $v$ $(v > 0)$ is random wind speed, $k$ is the Weibull shape parameter and $c$ is the Weibull scale parameter.

The two Weibull parameters are related by:

$$k = \left(\sqrt{D(v)}/E(v)\right)^{-1.086} \text{ and}$$ (2)

$$c = E(v)/\Gamma\left(1 + k^{-1}\right),$$ (3)

where $E(v)$ is the mean value of random wind speed $v$, $D(v)$ is the variance of random wind speed $v$ and $\Gamma()$ is the gamma function.

From Equation (1), the probability function of the Weibull distribution is given by:

$$F(v) = 1 - \exp^{-\left(\frac{v}{c}\right)^k}.$$ (4)

In Equation (4), it can be concluded that the probability function of the Weibull distribution *F(v)* is nondecreasing and ranges from 0 to 1. Therefore, the random wind speed $v$, subject to the probability function of Weibull distribution, is equivalent to the random number $m$ subject to a uniform distribution function ranging from 0 to 1. The random wind speed $v$ can be obtained by:

$$m = F(v) = 1 - \exp^{-\left(\frac{v}{c}\right)^k} \text{ and}$$ (5)

$$v = c[-\ln(1-m)]^{\frac{1}{k}}.$$ (6)

For example, it is known that *E(v)* is 11 m/s and *D(v)* is 5.16 m$^2$/s$^2$ for a wind field. According to Equations (2) and (3), it can be calculated that the shape parameter $k = 5.6$ and the scale parameter $c = 12$. Then, using Equation (6) and the Rand function in MATLAB software to generate the random number $m$, the time-history curve of the random wind-speed model is obtained by correlating the wind-speed data with the sampling points, as shown in Figure 1.

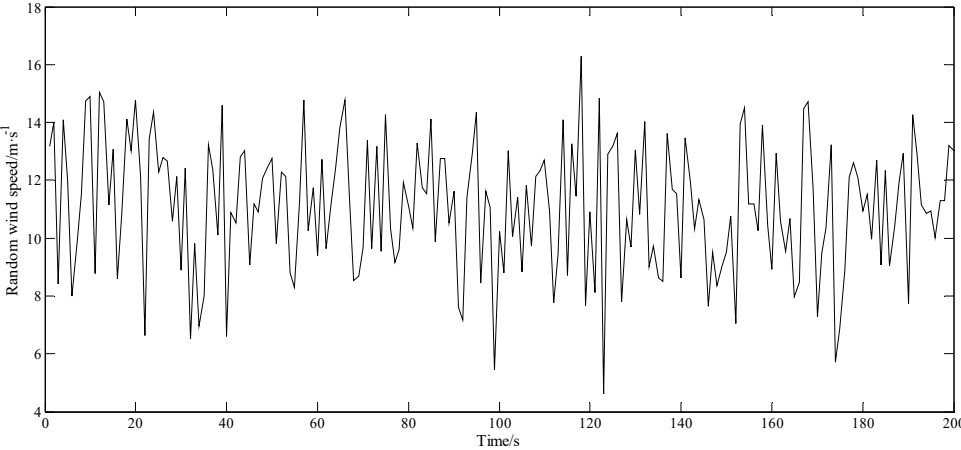

**Figure 1.** Time-history curve of a random wind-speed model.

## 3. Dynamics Model of the Multistage Transmission System

### 3.1. Basic Parameters of the Research Objective

The transmission system of a multistage gear train composed of a two-stage planetary gear train and a one-stage parallel-axis gear is shown in Figure 2. In Figure 2, 1r, 1p, 1s and 1c are the ring gear, planetary gear, sun gear and planetary carrier of the first-stage planetary gear train; 2r, 2p, 2s and 2c represent the corresponding units of the second-stage planetary gear train; 3g1 and 3g2 represent the driving gear and driven gear of the parallel-axis gear. The rated input power $P_{rate}$ of the transmission system is 2.8 MW, the rated input speed $n_{rate}$ is 18 rpm, the impeller radius $R$ is 50 m, the maximum output speed $n_{out}$ is 1887 rpm, and the transmission ratio $i$ is 104.867. The basic parameters of the transmission system are shown in Tables 1 and 2.

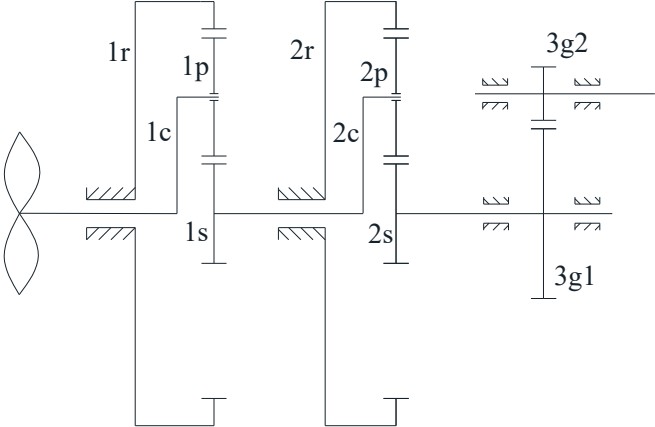

**Figure 2.** Structure diagram of the wind generator transmission system. 1r, 1p, 1s and 1c are the ring gear, planetary gear, sun gear and planetary carrier of first-stage planetary gear train. 2r, 2p, 2s and 2c represent the corresponding units of the second-stage planetary gear train; 3g1 and 3g2 represent the driving gear and driven gear of parallel-axis gear.

**Table 1.** Primary parameters of the planetary gear train.

|  | Parameter | Planetary Carrier | Ring Gear | Sun Gear | Planetary Gear |
|---|---|---|---|---|---|
| First-stage planetary gear train | Teeth number | - | 91 | 23 | 34 |
| | Module | - | 10 | 10 | 10 |
| | Pitch radius/mm | 285 | 455 | 115 | 170 |
| | Base circle radius/mm | - | 427.56 | 108.06 | 159.74 |
| | Mass/kg | 3240 | 448.96 | 918 | 756 |
| | Moment of inertia $J_1$/kg·m$^2$ | 680 | 65.4 | 35.4 | 26.8 |
| | Pressure angle $\alpha$/(°) | - | 20 | 20 | 20 |
| | Mesh stiffness $K_{sp}$, $K_{rp}$/N·m$^{-1}$ | - | $1.48 \times 10^{10}$ | $1.48 \times 10^{10}$ | $1.48 \times 10^{10}$ |
| | Coupling stiffness $K_{1s2c}$/N·m·rad$^{-1}$ | - | - | $7.53 \times 10^8$ | - |
| Second-stage planetary gear train | Teeth number | - | 121 | 23 | 49 |
| | Module | - | 10 | 10 | 10 |
| | Pitch radius/mm | 360 | 605 | 115 | 245 |
| | Base circle radius/mm | - | 568.51 | 108.06 | 230 |
| | Mass/kg | 3240 | 961 | 1069 | 895 |
| | Moment of inertia $J_1$/kg·m$^2$ | 680 | 245 | 42.9 | 33.4 |
| | Pressure angle $\alpha$/(°) | - | 20 | 20 | 20 |
| | Mesh stiffness $K_{sp}$, $K_{rp}$/N·m$^{-1}$ | - | $1.48 \times 10^{10}$ | $1.48 \times 10^{10}$ | $1.48 \times 10^{10}$ |
| | Coupling stiffness $K_{1s2c}$,$K_{2s3g1}$/N·m·rad$^{-1}$ | - | - | $7.53 \times 10^8$ | - |

**Table 2.** Primary parameters of the parallel-shaft gears.

| Parameter | Driving Gear | Driven Gear |
|---|---|---|
| Teeth number | 98 | 29 |
| Module | 6 | 6 |
| Pitch radius/mm | 294 | 87 |
| Base circle radius/mm | 276.27 | 81.75 |
| Mass/kg | 1096 | 102 |
| Moment of inertia $J_3$/kg·m$^2$ | 152 | 1.5 |
| Pressure angle $\alpha$/(°) | 20 | 20 |
| Mesh stiffness $K_{3g1g2}$/N·m$^{-1}$ | $1.48 \times 10^{10}$ | $1.48 \times 10^{10}$ |
| Coupling stiffness $K_{2s3g1}$/N·m·rad$^{-1}$ | $7.53 \times 10^8$ | - |

### 3.2. Dynamical Equations of the Transmission System

The power-flow route of the transmission system is: planetary carrier of first stage → sun gear of first stage → planetary carrier of second stage → sun gear of second stage → driving gear of parallel-shaft gears → driven gear of parallel-shaft gears. In order to facilitate the analysis and derivation, the dynamic model is appropriately simplified, and the following assumptions are made: each member is rigid, the planetary gears are evenly distributed around the circumference and the torsional stiffness of supporting members and the friction of the transmission system are ignored. Supposing that the meshing of gear teeth is simulated by spring and damper, a dynamic model (Figure 2) is established and shown in Figure 3, based on lumped parameter theory.

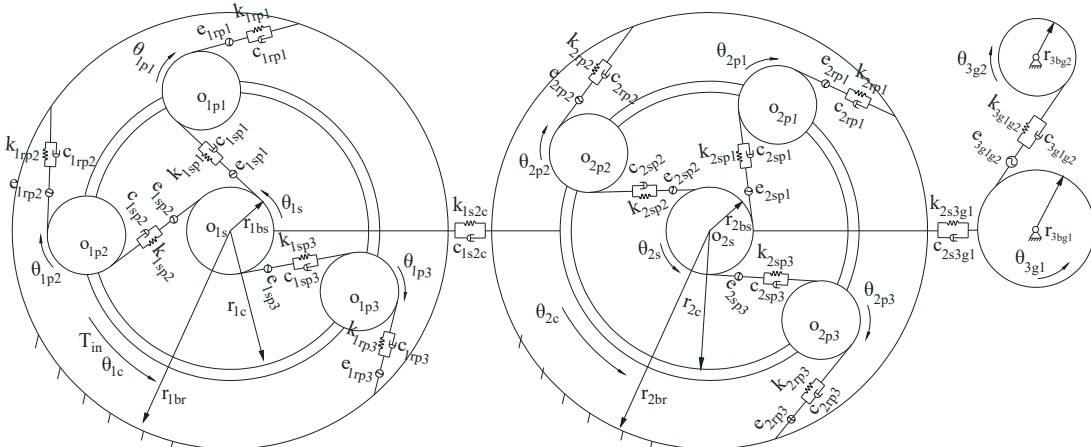

**Figure 3.** Dynamic model of the transmission system. $k_{1spj}$ ($j$ = 1,2,3) and $k_{1rpj}$ ($j$ = 1,2,3) are the mesh stiffness of the sun gear and planet gear, the ring gear and planet gear in the first-stage planetary gear train respectively, $c_{1spj}$ ($j$ = 1,2,3) and $c_{1rpj}$ ($j$ = 1,2,3) are the coupling stiffness of the sun gear and planet gear, the ring gear and planet gear respectively; $k_{2spj}$ ($j$ = 1,2,3) and $k_{2rpj}$ ($j$ = 1,2,3) are the mesh stiffness of the sun gear and planet gear, the ring gear and planet gear in the second-stage planetary gear train respectively. $c_{2spj}$ ($j$ = 1,2,3) and $c_{2rpj}$ ($j$ = 1,2,3) are the coupling stiffness of the sun gear and planet gear, the ring gear and planet gear, respectively.

The linear displacements of all components of the transmission system are shown in Equation (7):

$$\begin{cases} u_{1c} = r_{1c}\theta_{1c}, u_{1s} = r_{1bs}\theta_{1s} \\ u_{1pj} = r_{1bpj}\theta_{1pj}, u_{2c} = r_{2c}\theta_{2c} \\ u_{2s} = r_{2bs}\theta_{2s}, u_{2pj} = r_{2bpj}\theta_{2pj} \\ u_{3g1} = r_{3bg1}\theta_{3g1}, u_{3g2} = r_{3bg2}\theta_{3g2} \end{cases}, \tag{7}$$

where $\theta_{1i}$ ($i$ = s, c, r, $p_j$) is the angular displacement of each component in the first-stage planetary gear train around its rotation center and $\theta_{2i}$ is the angular displacement of each component in the second-stage planetary gear train around its rotation center.

Relative displacement of gear-meshing pairs of the transmission system along the meshing line are shown in Equation (8):

$$
\begin{cases}
\delta_{1spj} = u_{1s} + u_{1pj} - \cos\alpha_{1s} u_{1c} - e_{1spj} \\
\delta_{1rpj} = u_{1pj} - u_{1r} + \cos\alpha_{1r} u_{1c} - e_{1rpj} \\
\delta_{2spj} = u_{2s} + u_{2pj} - \cos\alpha_{2s} u_{2c} - e_{2spj} \\
\delta_{2rpj} = u_{2pj} - u_{2r} + \cos\alpha_{2r} u_{2c} - e_{2rpj} \\
\delta_{3g1g2} = u_{3g1} + u_{3g2} - e_{3g1g2}
\end{cases}, \tag{8}
$$

where $e_{1spj}$, $e_{1rpj}$ ($j$ = 1,2,3) is the meshing error of sun gear, ring gear and planetary gear in first-stage planetary gear train, respectively; $e_{2spj}$, $e_{2rpj}$ ($j$ = 1,2,3) is the meshing error of sun gear, ring gear and planetary gear in second-stage planetary gear train, respectively; $e_{3g1g2}$ is the meshing error of driving gear and driven gear in parallel-shaft gears.

By fixing the ring gear of the transmission system and taking the number of planet gears of the planetary gear train as three, namely 1N = 2N = 3, then according to the Lagrange general function, the dynamic equation of the transmission system shown in Figure 3 can be built, as indicated by Equation (9):

$$
\begin{cases}
m_{1c}\ddot{u}_{1c} - \cos\alpha_{1s}\sum_{j=1}^{3}\left(k_{1spj}\delta_{1spj} + c_{1spj}\dot{\delta}_{1spj}\right) + \cos\alpha_{1r}\sum_{j=1}^{3}\left(k_{1rpj}\delta_{1rpj} + c_{1rpj}\dot{\delta}_{1rpj}\right) = \frac{T_{in}}{r_{1c}} \\
m_{1s}\ddot{u}_{1s} + \sum_{j=1}^{3}\left(k_{1spj}\delta_{1spj} + c_{1spj}\dot{\delta}_{1spj}\right) + \frac{k_{1s2c}}{r_{1bs}}\left(\frac{u_{1s}}{r_{1bs}} - \frac{u_{2c}}{r_{2c}}\right) + \frac{c_{1s2c}}{r_{1bs}}\left(\frac{\dot{u}_{1s}}{r_{1bs}} - \frac{\dot{u}_{2c}}{r_{2c}}\right) = 0 \\
m_{1p1}\ddot{u}_{1p1} + k_{1sp1}\delta_{1sp1} + c_{1sp1}\dot{\delta}_{1sp1} + k_{1rp1}\delta_{1rp1} + c_{1rp1}\dot{\delta}_{1rp1} = 0 \\
m_{1p2}\ddot{u}_{1p2} + k_{1sp2}\delta_{1sp2} + c_{1sp2}\dot{\delta}_{1sp2} + k_{1rp2}\delta_{1rp2} + c_{1rp2}\dot{\delta}_{1rp2} = 0 \\
m_{1p3}\ddot{u}_{1p3} + k_{1sp3}\delta_{1sp3} + c_{1sp3}\dot{\delta}_{1sp3} + k_{1rp3}\delta_{1rp3} + c_{1rp3}\dot{\delta}_{1rp3} = 0 \\
m_{2c}\ddot{u}_{2c} - \cos\alpha_{2s}\sum_{j=1}^{3}\left(k_{2spj}\delta_{2spj} + c_{2spj}\dot{\delta}_{2spj}\right) + \cos\alpha_{2r}\sum_{j=1}^{3}\left(k_{2rpj}\delta_{2rpj} + c_{2rpj}\dot{\delta}_{2rpj}\right) - \frac{k_{1s2c}}{r_{2c}}\left(\frac{u_{1s}}{r_{1bs}} - \frac{u_{2c}}{r_{2c}}\right) - \frac{c_{1s2c}}{r_{2c}}\left(\frac{\dot{u}_{1s}}{r_{1bs}} - \frac{\dot{u}_{2c}}{r_{2c}}\right) = 0 \\
m_{2s}\ddot{u}_{2s} + \sum_{j=1}^{3}\left(k_{2spj}\delta_{2spj} + c_{2spj}\dot{\delta}_{2spj}\right) + \frac{k_{2s3g1}}{r_{2bs}}\left(\frac{u_{2s}}{r_{2bs}} - \frac{u_{3g1}}{r_{3bg1}}\right) + \frac{c_{2s3g1}}{r_{2bs}}\left(\frac{\dot{u}_{2s}}{r_{2bs}} - \frac{\dot{u}_{3g1}}{r_{3bg1}}\right) = 0 \\
m_{2p1}\ddot{u}_{2p1} + k_{2sp1}\delta_{2sp1} + c_{2sp1}\dot{\delta}_{2sp1} + k_{2rp1}\delta_{2rp1} + c_{2rp1}\dot{\delta}_{2rp1} = 0 \\
m_{2p2}\ddot{u}_{2p2} + k_{2sp2}\delta_{2sp2} + c_{2sp2}\dot{\delta}_{2sp2} + k_{2rp2}\delta_{2rp2} + c_{2rp2}\dot{\delta}_{2rp2} = 0 \\
m_{2p3}\ddot{u}_{2p3} + k_{2sp3}\delta_{2sp3} + c_{2sp3}\dot{\delta}_{2sp3} + k_{2rp3}\delta_{2rp3} + c_{2rp3}\dot{\delta}_{2rp3} = 0 \\
m_{3g1}\ddot{u}_{3g1} + k_{3g1g2}\delta_{3g1g2} + c_{3g1g2}\dot{\delta}_{3g1g2} - \frac{k_{2s3g1}}{r_{3bg1}}\left(\frac{u_{2s}}{r_{2bs}} - \frac{u_{3g1}}{r_{3bg1}}\right) - \frac{c_{2s3g1}}{r_{3bg1}}\left(\frac{\dot{u}_{2s}}{r_{2bs}} - \frac{\dot{u}_{3g1}}{r_{3bg1}}\right) = 0 \\
m_{3g2}\ddot{u}_{3g2} + k_{3g1g2}\delta_{3g1g2} + c_{3g1g2}\dot{\delta}_{3g1g2} = \frac{-T_{out}}{r_{3bg1}}
\end{cases}. \tag{9}
$$

The dynamic equation of the transmission system is converted into matrix form as:

$$
M\ddot{x} + Kx + C\dot{x} = F. \tag{10}
$$

There are 12 degrees of freedom in the matrices given in Equation (10), where $M$ is the mass matrix, $K$ is the stiffness matrix, $C$ is the damping matrix, $F$ is the load vector and $x$ is the displacement vector, as $x = [u_{1c}, u_{1s}, u_{1p1}, u_{1p2}, u_{1p3}, u_{2c}, u_{2s}, u_{2p1}, u_{2p2}, u_{2p3}, u_{3g1}, u_{3g2}]^T$.

## 4. Analysis of Dynamic Load-Sharing Performance

### 4.1. External Excitation of the Transmission System

Relevant data are provided by a wind farm: the cut-in wind speed $v_{cutin}$ is 3 m/s, the cut-out wind speed $v_{cutoff}$ is 25 m/s, the rated wind speed $v_{rate}$ is 12 m/s, wind density $\rho$ is 1.21 kg/m$^3$ and the wind-energy-utilization coefficient $c_p$ is 0.55. Based on the relationship between the input speed $n_{in}$ of

the transmission system and the random wind speed $v$, as shown in Equation (11), the input speed curve of the transmission system in the first 100 s is established, as shown in Figure 4.

$$
n_{\text{in}} = \begin{cases} 0, & v < v_{cut\,in} \\ \frac{n_{\text{rate}}}{v_{rate}}v, & v_{cut\,in} \le v < v_{rate} \\ n_{\text{rate}}, & v_{rate} \le v \le v_{cut\,off} \\ 0, & v_{cut\,off} > v \end{cases}. \tag{11}
$$

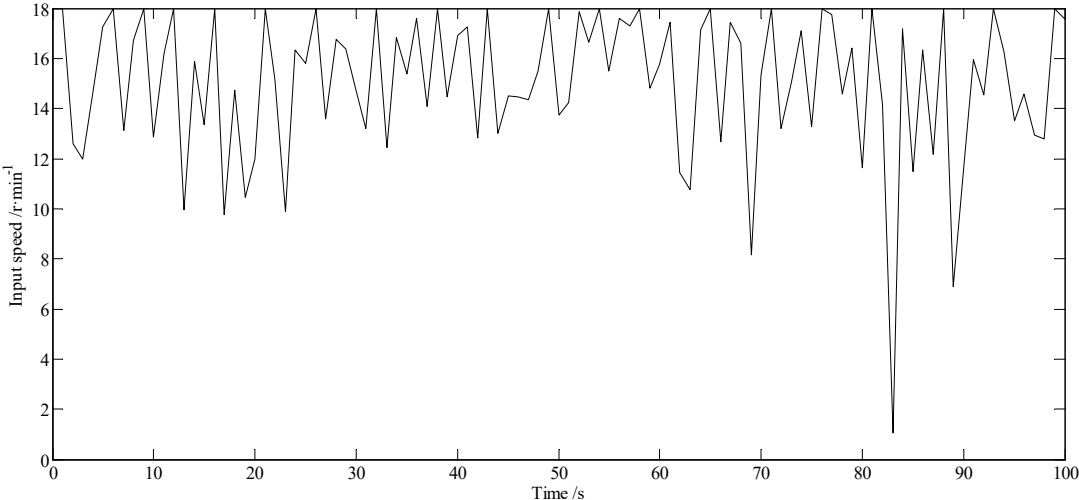

**Figure 4.** Input speed curve of the transmission system in 100 s.

### 4.2. Calculation of Mesh-Load Factor

For Equation (9) and at equilibrium, the dynamic force between the planet gear and sun gear is $F_{spj}$, and $F_{rpj}$ is the dynamic force between the planet gear and ring gear. The two forces are given as:

$$
\begin{cases} F_{spj} = k_{spj}\delta_{spj} + c_{spj}\dot{\delta}_{spj} \\ F_{rpj} = k_{rpj}\delta_{rpj} + c_{rpj}\dot{\delta}_{rpj} \end{cases}. \tag{12}
$$

A numerical integration method is used to solve Equation (10), which represents the dynamic transmission system, by making $D_{1spjks}$ and $D_{1rpjkr}$ represent the load-sharing coefficients of the external and internal meshing, respectively, of all gear pairs in the first-stage planetary gear train; $D_{2spjks}$ and $D_{2rpjkr}$ represent the load-sharing coefficients of the external and internal meshing, respectively of all gear pairs in the second-stage planetary gear train, as shown in Equation (13):

$$
\begin{cases} D_{1spjks} = \dfrac{1N(F_{1spjks})_{\max}}{\sum\limits_{j=1}^{1N}(F_{1spjks})_{\max}}, & D_{1rpjkr} = \dfrac{1N(F_{1rpjkr})_{\max}}{\sum\limits_{j=1}^{1N}(F_{1rpjkr})_{\max}} \\ D_{2spjks} = \dfrac{2N(F_{2spjks})_{\max}}{\sum\limits_{j=1}^{2N}(F_{2spjks})_{\max}}, & D_{2rpjkr} = \dfrac{2N(F_{2rpjkr})_{\max}}{\sum\limits_{j=1}^{2N}(F_{2rpjkr})_{\max}} \end{cases}, \tag{13}
$$

where $ks$ is the meshing cycle number for the external meshing of the planetary gear pair and $kr$ is the meshing cycle number for the internal meshing of the planetary gear pair.

Because the wind generator is often affected by irregular wind force, in order to more accurately reflect the load-sharing characteristic of each planet gear pair of the transmission system under the excitation of random wind speed, 100 points of input speed were randomly selected from Figure 4 as the external excitation and the fluctuation diagram of random input speed, as shown in Figure 5.

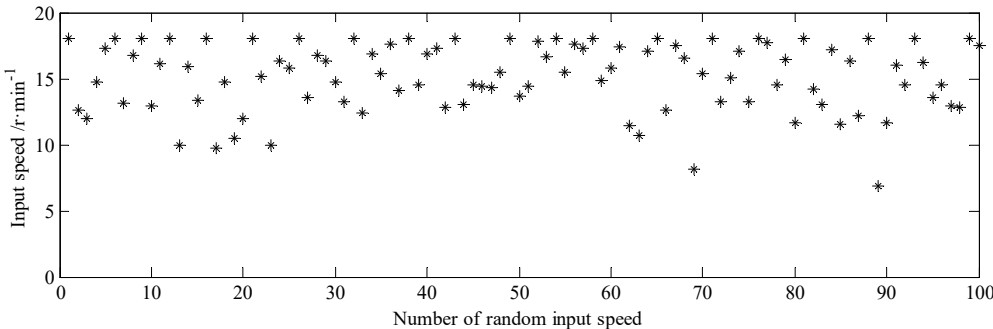

**Figure 5.** The fluctuation diagram of 100 points random input speed.

The results of substituting the relevant parameters of the transmission system into Equation (10) for solution and using Equation (13) to obtain the dynamic load-sharing coefficients of each planetary gear pair in the same meshing period are shown in Figures 6 and 7.

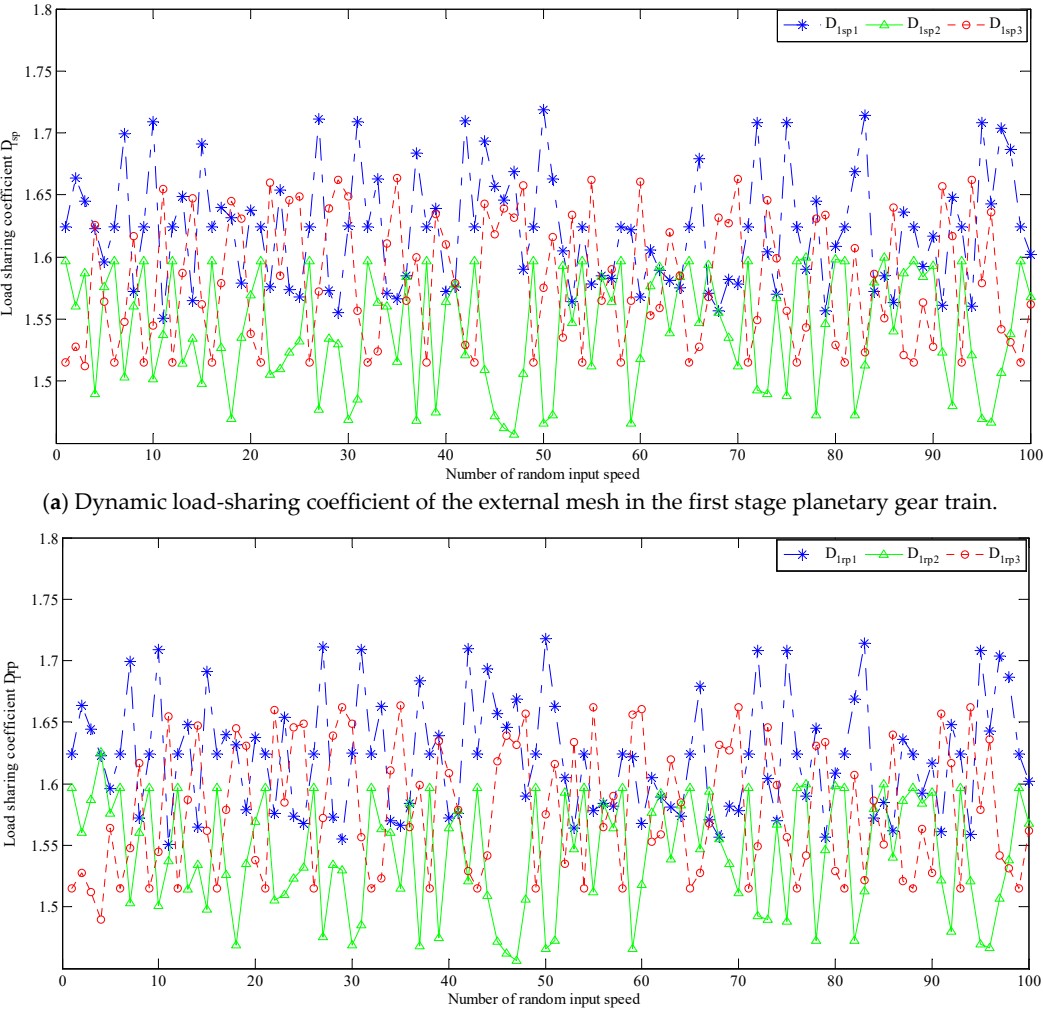

(**a**) Dynamic load-sharing coefficient of the external mesh in the first stage planetary gear train.

(**b**) Dynamic load-sharing coefficient of the internal mesh in the first stage planetary gear train.

**Figure 6.** Dynamic load-sharing coefficient of the first-stage planetary gear train under random input wind speed. (**a**) Dynamic load-sharing coefficient of the external mesh in the first-stage planetary gear train. (**b**) Dynamic load-sharing coefficient of the internal mesh in the first-stage planetary gear train. $D_{1sp1}$, $D_{1sp2}$, and $D_{1sp3}$ are the dynamic load-sharing coefficient of each external planet gear pair of the first-stage planetary gear train, respectively. $D_{1rp1}$, $D_{1rp2}$ and $D_{1rp3}$ are the dynamic load-sharing coefficient of each internal planet gear pair respectively.

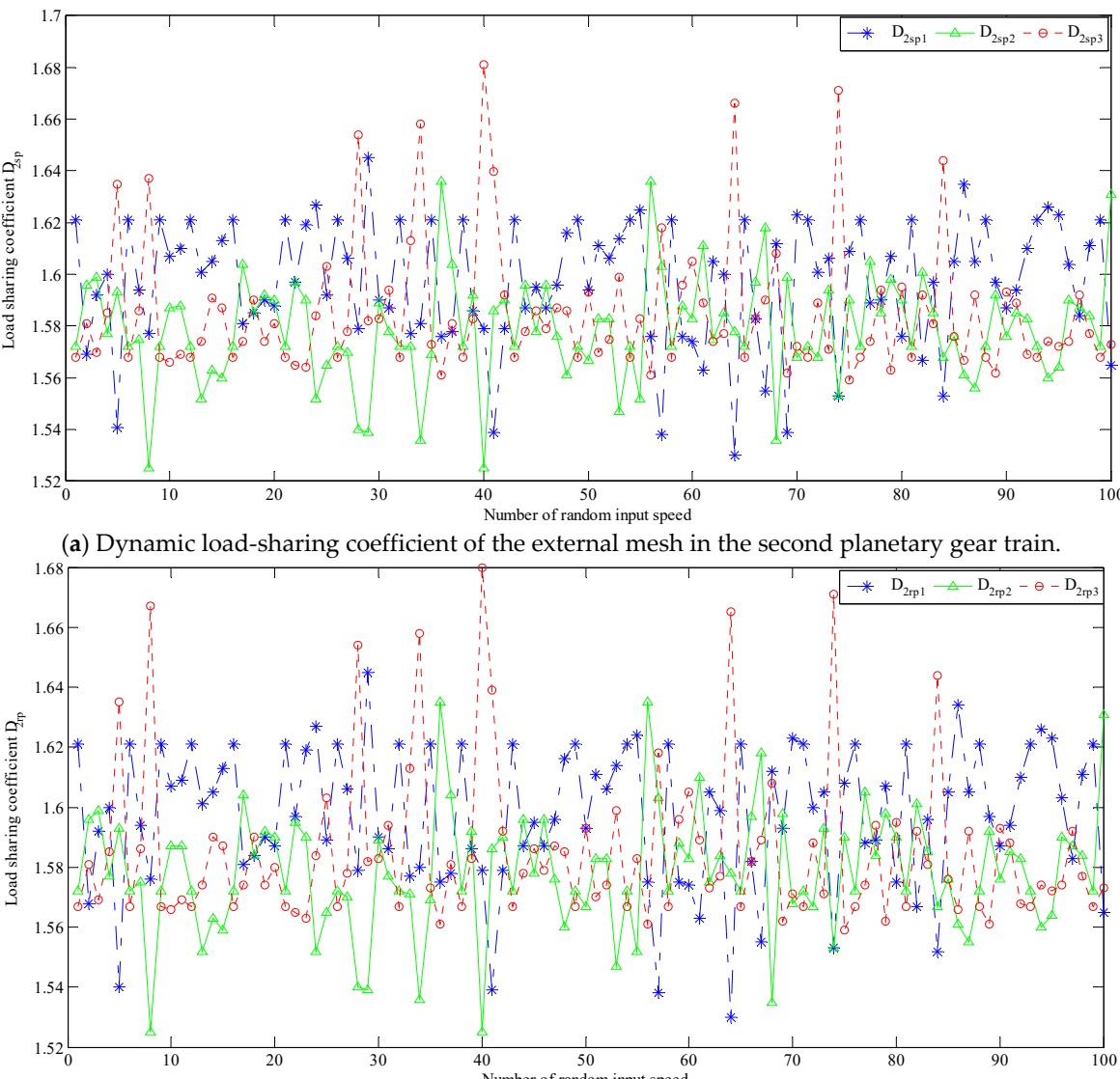

(**a**) Dynamic load-sharing coefficient of the external mesh in the second planetary gear train.

(**b**) Dynamic load-sharing coefficient of the internal mesh in the second planetary gear train.

**Figure 7.** Dynamic load-sharing coefficient of the second planetary gear train under random input wind speed. (**a**) Dynamic load-sharing coefficient of the external mesh in the second planetary gear train. (**b**) Dynamic load-sharing coefficient of the internal mesh in the second planetary gear train. $D_{2sp1}$, $D_{2sp2}$, and $D_{2sp3}$ are the dynamic load-sharing coefficient of each external planet gear pair of the second-stage planetary gear train, respectively. $D_{2rp1}$, $D_{2rp2}$ and $D_{2rp3}$ are the dynamic load-sharing coefficient of each internal planet gear pair respectively.

In order to further analyze the fluctuation degree of load-sharing coefficient for each planet gear pair of the transmission system, the relationship between the load-sharing coefficient curve of external meshing of the first- and second-stage planetary gear train is drawn in Figure 8.

Through data analysis, the fluctuation degree of the load-sharing coefficient of each planet gear pair in the transmission system at 100 points random input speed can be calculated, as shown in Figure 9.

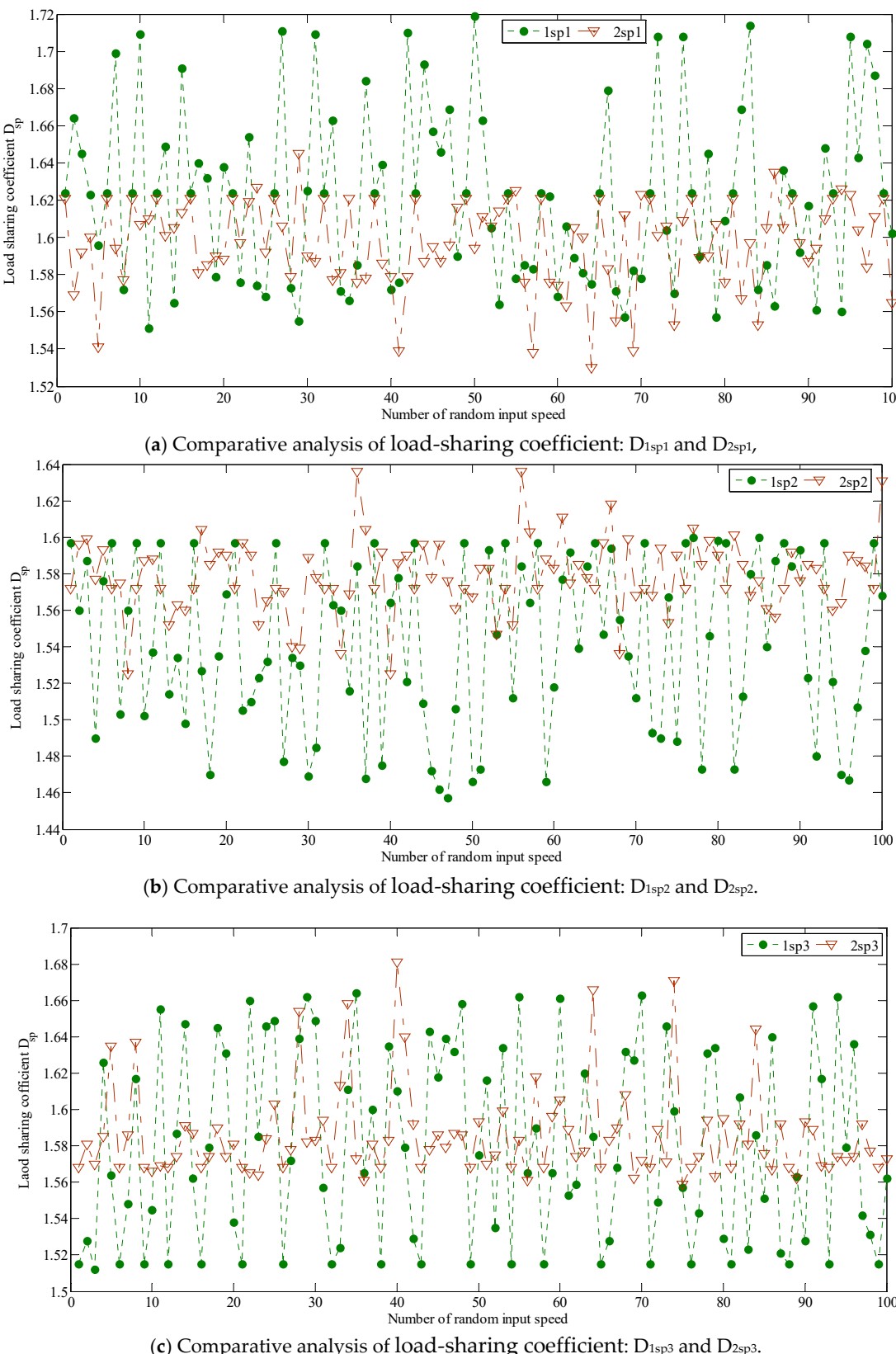

(**a**) Comparative analysis of load-sharing coefficient: $D_{1sp1}$ and $D_{2sp1}$,

(**b**) Comparative analysis of load-sharing coefficient: $D_{1sp2}$ and $D_{2sp2}$.

(**c**) Comparative analysis of load-sharing coefficient: $D_{1sp3}$ and $D_{2sp3}$.

**Figure 8.** Load-sharing coefficient fluctuation degree curve of the external meshing of the first stage and second stage planetary gear train. (**a**) Comparative analysis of load-sharing coefficient: $D_{1sp1}$ and $D_{2sp1}$. (**b**) Comparative analysis of load-sharing coefficient: $D_{1sp2}$ and $D_{2sp2}$. (**c**) Comparative analysis of load-sharing coefficient: $D_{1sp3}$ and $D_{2sp3}$.

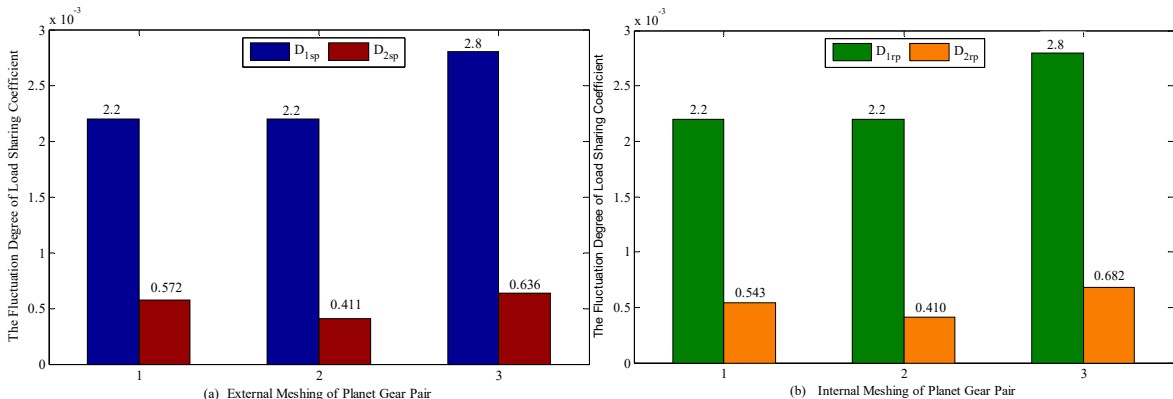

**Figure 9.** Data analysis of the load-sharing coefficient fluctuation degree of each planet gear pair in the transmission system. (**a**) External meshing of the planet gear pair. (**b**) Internal meshing of the planet gear pair.

Here, the variance ($s^2$) is used to reflect the load-sharing coefficient fluctuation degree of each planet gear pair in the same meshing period of the transmission system under random input speed. Figure 9 indicates that the fluctuation degree of the load-sharing coefficient of each planet gear pair in the first stage planetary gear train is: $s^2_{(D1sp1)} = 2.2 \times 10^{-3}$, $s^2_{(D1sp2)} = 2.2 \times 10^{-3}$, $s^2_{(D1sp3)} = 2.8 \times 10^{-3}$, $s^2_{(D1rp1)} = 2.2 \times 10^{-3}$, $s^2_{(D1rp2)} = 2.2 \times 10^{-3}$, $s^2_{(D1rp3)} = 2.8 \times 10^{-3}$; the fluctuation degree of the load-sharing coefficient of each planet gear pair in the second stage planetary gear train is: $s^2_{(D2sp1)} = 5.72 \times 10^{-4}$, $s^2_{(D2sp2)} = 4.11 \times 10^{-4}$, $s^2_{(D2sp3)} = 6.36 \times 10^{-4}$, $s^2_{(D2rp1)} = 5.43 \times 10^{-4}$, $s^2_{(D2rp2)} = 4.1 \times 10^{-4}$, $s^2_{(D2rp3)} = 6.82 \times 10^{-4}$. By comparing and analyzing, the fluctuation degree of the load-sharing coefficient of each planet gear pair in the first-stage planetary gear train is greater than that of each planet gear pair in the second-stage planetary gear train. Figures 6–8 show that the fluctuation degree of the load-sharing coefficient of each planetary gear pair in the first-stage planetary gear train is more obvious.

## 5. Conclusions

A dynamics model for a multistage planetary gear transmission system has been established to study the load-sharing performance of each planet gear pair in the same meshing period of a system which is under the interaction of random input speed and internal excitation. The results show that the fluctuation degree of the load-sharing coefficient of each planet gear pair in the first-stage planetary gear train is more intense, and it is necessary to improve load distribution of gear pairs in the first-stage planetary gear train, which reduces vibration and increases the life of the wind turbine generator.

**Author Contributions:** In this research contribution, J.W. is involved in system modelling, simulation and results analysis; Y.L. supervised the progress of research; S.Y. and R.M. wrote the paper.

**Funding:** This research was funded by National Natural Science Foundation of China, Grant number: 51765020.

**Acknowledgments:** The authors would like to thank anonymous reviewers for their helpful comments and suggestions to improve the manuscript. This project is supported by National Natural Science Foundation of China (Grant No.51765020).

**Conflicts of Interest:** The authors declare no conflict of interest.

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
