# Peer review of "Analysis of Load-Sharing Behavior of the Multistage Planetary Gear Train Used in Wind Generators: Effects of Random Wind Load"

_applsci, doi:10.3390/app9245501_

Round 1

Reviewer 1 Report

This paper is analyzed load sharing behavior of multistage planetary gear-train used in wind generator considering random wind load. It is expected that the suggested method has the benefit of analysis of load sharing of gear-train. However, it is difficult to confirm the usefulness of this paper due to the lack of the motivation of this study, the validity of the random wind speed model, information of gear-train, and the statistical analysis for load sharing results. The detail comments were listed below:

In this study, lots of related studies were referred. The authors mentioned that the originality of this paper is the input of random loads, wind generator, and so on. However, the originality of this paper should focus on the method of analysis rather than on the object because there are so many different objects with various load conditions.

The author used only 1 random wind speed model such as Fig. 1. The author should not use 1 random load but lots of random load because the objective of this study is load sharing characteristic of multistage planetary gear transmission under random wind load condition.

On the dynamics model of the multistage transmission system paragraph, the author should explain the multistage transmission system used in this paper. But, the author explained very simple structure diagram of the wind generator. So, it needs to explain whole system of this power flow in detail.

The results showed that the fluctuation degree of load sharing coefficient of each gear pair in the first stage planetary gear is more obvious. I understand this result in Figure 7. However, it is very vague to explain it in graphs. Therefore, it is necessary to describe the conclusion through statistical analysis so that the reader can easily understand.

Reviewer 2 Report

I suggest to use standard terminology (AGMA, ISO):

Tooth number – teeth number (number of teeth)

Modulus – module

Inner ring – ring gear  

loading sharing coefficient – mesh load factor (ISO 6336 and AGMA 6123-B06);

Standard description is mesh stiffness but meshing stiffness is understandable too

Mistakes:

In Table 1 for third stage what is wrong – the module (6 mm) or the radii?

In Table 2: In 17th  row, last column 1.48∙1010 must be 1.48∙1010

In PDF format all points above the first and second derivatives are not visible.

Suggestions:

Is the investigated gearbox a real, manufactured and working arrangement or only authors proposal?

In first case the manufacturer must be cited, I think. In second case the authors must write a few words about:

Why the gear wheels of the first and the second stage are with equal face width? Usually in equal module the face widths or planet numbers are different because different mesh load. Of course there are cases of equal face width but this circumstance must be underlined in the text. If the face width is different, why the meshing stiffness is equal?

It is not clear the type of connection between 1s and 2c (Fig. 2) and 2s and 3g1. This is important for determination of masses of sun gears and floating possibilities of members (which reflects to the dynamic model). I fill a lack of a design drawing.

In case of constant input torque there is uneven load distribution between planets considered by load mesh factor Kγ (see ISO 6336). This unevenness depends on many factors (listed in AGMA 6123-B06 and described in Arnaudov at al.). Some of them are ignored in dynamic model (Fig. 3) – which? The dynamic model in Fig. 3 is simplified and I suggest to list main simplifying prepositions (no equalizing devices, misalignments of planet centers positioning, difference in planets bearings clearance, etc.).

It is not clear in the paper how calculated load sharing coefficient D (formula 13) relates to Kγ. Maybe the short way to show this is present diagram of D (similar to Fig. 5) in case of no input speed fluctuation (or mention in the text).

In formulae (13) one symbol is used for different variables. In the denominator (F1spjks)max is the maximal force in each of planets j = 1, 2, and 3. This means three forces (F1sp1ks)max, (F1sp2ks)max, and (F1sp3ks)max. In the numerator the same symbol (F1spjks)max means one force – the maximal of above mentioned three forces. Maybe this is not very clear.

Mentioned sources (this is not suggestion to cite them)

Arnaudov, K. and D. Karaivanov. Planetary gear trains. Boca Raton [FL, USA]: CRC Press, 2019, 358 p., ISBN 978-1-138-31185-5.

Arnaudov, K., D. Karaivanov, and L. Dimitrov. Some Practical Problems of Distribution and Equalization of Internal Loads in Planetary Gear Trains. Mechanisms and Machine Science, 13 (2013), Power Transmissions, Proceedings of the 4th International Conference, Sinaia, Romania, June 20-23, 2012, Editor Georg Dobre, Springer Dordrecht Heidelberg New York London, p. 585-596. ISSN 2211-0984.

Round 2

Reviewer 1 Report

Most of the manuscripts was modified according to the review comments. Only 1 comments should be modified.

“4. Comment: The results showed that the fluctuation degree of load sharing coefficient of each gear pair in the first stage planetary gear is more obvious. It is very vague to explain it in graphs. Therefore, it is necessary to describe the conclusion through statistical analysis so that the reader can easily understand. Response to comment: Considering the Reviewer’s suggestion, the magnitude of the load sharing coefficients of each planetary gear pair in the same meshing period under 100 points random input speed, as shown in appendix. The fluctuation degree of the load sharing coefficient of each planet gear pair in the transmission system is calculated and analyzed.”

But, the author must present the statistical results regarding the fluctuation degree of load sharing coefficient of each gear pair. The author presented the appendix. I understand it by the figure 6, 7, and 8 as well as the appendix. I also know that it can be seen that the fluctuation degree of the load sharing coefficient of each planetary gear pair in the first stage planetary gear train is more obvious.  However, the author needs to quantify how much larger it is. Therefore, it is necessary to discuss the results using statistical methods. Also, this difference should discuss how effective the method presented in this study is.
